



# Size distribution law of earthquake-triggered landslides in different seismic intensity zones

Yidan Huang[1,2,3], Lingkan Yao[1,2,3]

[1] School of Civil Engineering, Southwest Jiaotong University, Chengdu, 610031, China
[2] MOE Key Laboratory of High-speed Railway Engineering, Chengdu 610031, China
[3] Road and Railway Engineering Research Institute, Sichuan Key Laboratory of Seismic Engineering and Technology, Chengdu 610031, China

*Correspondence to*: Lingkan Yao (yaolk@swjtu.edu.cn)

**Abstract.** The Ms 8.0 Wenchuan earthquake in 2008 and Ms 7.0 Lushan earthquake in 2013 produced thousands of landslides in the southern region of the Longmen Mountains in China. We conducted field investigations and analyzed remote sensing data to determine the distribution law of earthquake-triggered landslides. The results show a strong negative power-law relationship between the size and frequency of landslides in VII, VIII, and IX seismic intensity zones, a weak power law in the X seismic intensity zone, and a lognormal distribution in the XI seismic intensity zone. Landslide density
increases with increasing seismic intensity. A sand pile cellular automata model was built under the conceptual framework of self-organized criticality theory to simulate earthquake-induced landslides. Data from the simulations demonstrate that with increasing disturbance intensity, the dynamical mechanism of the sand pile model changes from a strong power law to a weak power law and then to a lognormal distribution. Results from shaking table experiments of a one-sided slope sand pile show that for peak ground acceleration (PGA) in the range of 0.075 g–0.125 g, the relation between the amount and
frequency of sand follows a negative power law. For PGA between 0.15 g and 0.25 g, the relation obeys a lognormal distribution. This verifies that the above-mentioned distribution of earthquake-induced landslides should be a universal law from a physical viewpoint and may apply to other areas. This new perspective may be used to guide development of an inventory of earthquake-triggered landslides and provide a scientific basis for their prediction.

## 1 Introduction

Earthquakes have triggered numerous landslides. The 1994 Mw 6.7 Northridge earthquake in the USA triggered more than 11,000 landslides (Harp and Jibson, 1995, 1996). The 2008 Mw 7.9 Wenchuan earthquake in China triggered 197,481 landslides (Xu et al., 2014). The 2014 Mw 7.0 Port-au-Prince Earthquake in Haiti triggered 30,828 landslides (Xu et al.,





2014). Landslides are one of the most common and dangerous coseismic disasters. In particular, landslides triggered by the

2008 Wenchuan earthquake induced about one-third of total deaths and disappearances (Huang and Li, 2009).

An inventory map of earthquake-induced landslides shows many correlations between landslides and seismic factors (e.g., distance from the epicenter and major surface rupture, seismic intensity, peak ground acceleration, slope gradient), slope characteristic factors (e.g., gradient, aspect, elevation, lithology,), and have been analyzed using statistical methods (Keefer, 2000; Rodriguez et al., 1999; Papadopoulos, 2000; Parise and Jibson, 2000; Meunier et al., 2007; Harp et al., 2011; Guo et

al., 2017). Many significant laws, such as the effects of slope aspect, faults, hanging wall, and topography, have been reported (Huang and Li, 2009; Yin et al., 2009; Dai et al., 2011; Meunier et al., 2008). Most studies address the relationship between landslide frequency and magnitude associated with a trigger (e.g., earthquake) or historical landslide inventory (Malamud et al., 2004). These studies used statistical models including the Pareto probability model (Frattini and Crosta, 2013), double Pareto probability model (Stark and Hovius, 2001), and gamma function probability model (Van et al., 2007).

The frequency-magnitude distribution of landslides is typically expressed in a non-cumulative (Carrara et al., 2003; Guzzetti et al., 2002) and cumulative form (Hovius et al., 1997; Guthrie and Evans, 2004). Landslide magnitude is often expressed in terms of area (Catani et al., 2005; Havenith et al., 2006; Chien et al., 2007) or volume (Brunetti et al., 2009; Jaiswal et al., 2012), both of which are representative of landslide magnitude and have a certain correlation. Area is widely used because it is easier to obtain than volume.

Several studies have shown that the frequency-magnitude distribution of landslides follows a power-law distribution (Frattini and Grosta, 2013; Pelletier et al., 1997; Malamud and Turcotte, 1999; Iwahashi et al., 2003; Guthrie and Evans, 2004; Dahl et al., 2013; Harp and Jibson, 1996). Most studies, however, focused on phenomena rather than causes. The size distribution law of landslides and its physical mechanism remain poorly understood. Hergarten (1998, 2003) compared observed landslide size distributions with models of self-organized criticality (SOC) and suggested that landsliding in combination

with its driving processes may be considered a SOC phenomenon. Unfortunately, no further work was undertaken owing to the lack of available field data.

Self-organized criticality theory belongs to the field of nonlinear physics field and is a new concept to explain the behavior of composite systems developed by Bak et al. (1987, 1988). This kind of system or composite system contains an exceedingly large number of elements that interact over a short range, naturally evolve toward a critical state, and locked in

this state over a given period of time, or permanently. In the critical state, a minor event can initiate a chain reaction that affects a large number of elements in the system and even lead to a catastrophe. Although systems produce more minor events than catastrophes, chain reactions of all sizes are an integral part of the dynamics. According to theory, the mechanism that leads to minor events is the same that leads to major events.





A sand pile is a deceptively simple system that serves as a paradigm for SOC. Held and co-workers (1990) devised a
precision apparatus that slowly and uniformly pours sand grain by grain onto a flat, circular surface. The sand pile stops
growing when the amount of sand added is balanced on average by the amount of sand that falls off the edge, at which point
the system has reached a critical state. When a grain of sand is added to a pile in the critical state, it can initiate an avalanche
of any size, including a catastrophic event, and the size and the frequency are related by a power law. SOC has explained the
dynamics of many catastrophes such as earthquakes (Olami et al., 1992), forest fires (Drossel and Schwabl ., 1992; Malamud
et al., 1998), mountain and rock slides, snow avalanches (Hergarten, 2002,2003), solar and stellar flares (Aschwanden,
2011a, 2016), and stock market crashes (Sornette, 2003). This paper investigates SOC's application to landslides triggered
by earthquakes.

Two great earthquakes struck the Longmen Mountains in 2008 and 2013 and triggered numerous landslides. The Longmen
Mountains, on the eastern edge of the Qinghai-Tibetan Plateau in Sichuan Province, China, have the greatest gradient change
in the surrounding mountain system. At least 5–10 km of material in the Longmen Mountains has been eroded away since
the Miocene, rising at a rate of about 0.6 mm/y (Li Yong et al., 2006; Liu Shugen, 1993). The continuous effects of uplifting
and denudation formed a deep and steep valley landscape with an average slope in the southern and Minjiang River areas
above 27°. According to Davis's theory of erosion cycles (Davis, 1899), the Longmen Mountains are in the early maturity of
geomorphologic evolution and their slopes have evolved to a critical stage. Common characteristic of slope disasters include
dilapidation of natural terrain or cut slope, rock avalanches, debris flows, landslides, rock piles, and snowslides, all of which
are mainly composed of granular mixtures and associated with energy dissipation owing to material instability and slipping
during slope accumulation. The sand pile model is ideal for reflecting the energy dissipation process within slope
accumulation, and its dynamic characteristics should therefore be explainable under the conceptual framework of SOC ( Yao,
et al., 2003; Yan and Huang, 2016).

In this study, we aim to better understand the universal behavior of slope disasters from the point of view of SOC. We
studied the distribution of landslides induced by two earthquakes in the Longmen Mountains by field investigations and
remote sensing interpretation. A sand pile cellular automata model was also built under the conceptual framework of SOC to
simulate earthquake-induced landslides. Shaking table experiments of a one-sided slope sand pile under seismic excitations
were conducted. The results of this study provide a physical explanation of the general distribution of earthquake-induced
landslides.





## 2 Landslide magnitude-frequency distributions in different seismic intensity zones

### 2.1 Study area

The 2008 Wenchuan earthquake triggered the largest number of landslides observed in human history. On April 20, 2013, the Lushan earthquake also triggered a large number of landslides. Both events were caused by thrust faulting in the Longmen Mountains of China in areas under similar geographical, geological, and geomorphological conditions.

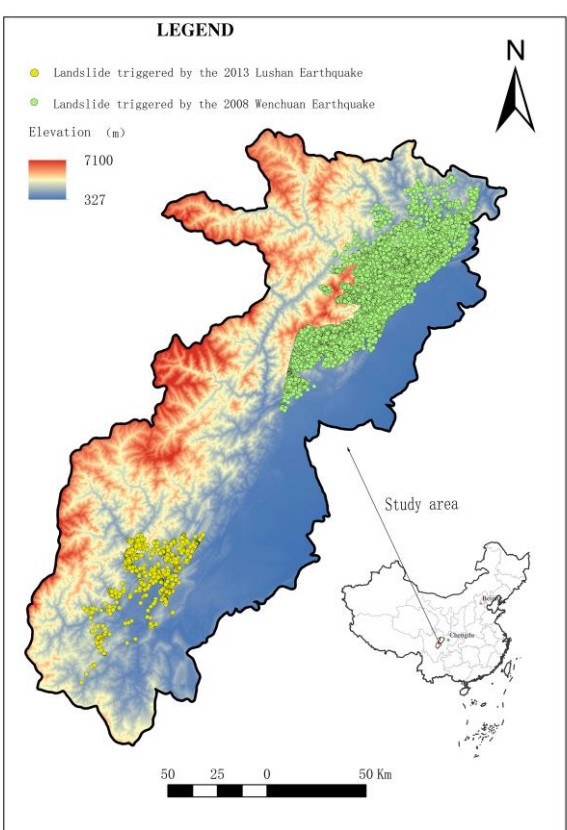

**Figure 1: Landslides triggered by the 2008 Wenchuan and 2013 Lushan earthquakes**

### 2.2 Data and Methodology

#### 2.2.1 Landslide field investigation

The majority of our field work was accessed by car, landslides were recorded that were visible along the road, and then followed by foot for detailed investigation. The landslide investigation of the Wenchuan earthquake began in the rescue time



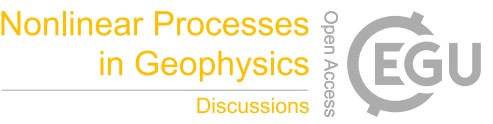

from May 31, 2008 soon after the main shock until June 5, 2008. We collected preliminary survey data by recording the location of the landslide deposits that covered the road, estimating the landslide volume, and asking the highway department staff for the amount of cleaned landslide deposits. We chose to limit the survey to >10 m$^3$ landslides that blocked roads.

A relatively detailed reconnaissance field study was conducted from August 2 to 27, 2009. After two rainy seasons, the loose material on the slope was nearly washed away, revealing a complete sliding bed or collapsed back wall. We measured the spatial position coordinates of the sliding bed and compared with a 1:2000 topographic map along the highway from Dujiangyan to Yingxiu, which is a section of Chinese national highway 213. A three-dimensional digital figure of the landslide was generated to obtain the landslide volume and depth. According to the slope engineering design scheme of the

highway, the landslide depth can be estimated more accurately by measuring the length of exposed anchors after the earthquake.

After the Lushan earthquake, we conducted landslide field surveys along the road from April 26 to August 26, 2013. We accumulated survey routes exceeding 500 km in length, including the China national highway 318, provincial highways 210 and 211, country town roads from Leyin to Rongjin, Longmen to Taiping, Renjia to Shuangshi, Shuangshi to Linguan, and

Taiping to Dachuan. We focus on landslide volume, which is mainly obtained from field measurements and the amount of cleaned landslide deposits by the highway department during the rescue time.

### 2.2.2 Landslide inventory mapping

Landslides triggered by earthquake are usually large in number and widespread in distribution, and detailed coseismic landslide data cannot be based on field investigations alone. At present, the visual interpretation of high-resolution remote

sensing images is the primary method to obtain large-area earthquake-induced landslide data.

We collected pre- and post-earthquake high-resolution remote sensing data to facilitate the landslide analysis. The Wenchuan earthquake data included TM satellite images (pre-earthquake), ALOS satellite images (10-m spatial resolution, taken on June 4, 2008), and Quick-Bird satellite images (0.61-m spatial resolution, taken on May 30, 2008). The Lushan earthquake data included LANDSAT-5, SPOT4, and SPOT4 satellite images (pre-earthquake), aerial photographs (0.4-m and 2-m

spatial resolution, taken on April 20 and 21, 2013) provided by the Institute of Remote Sensing and Digital Earth of the Chinese Academy of Sciences, aerial photographs (Baoshen, Longmen, and Taipin towns, 0.16-m spatial resolution, taken on April 20, 2013) provided by the State Bureau of Surveying and Mapping, and ZY3 satellite images (2.1-m spatial resolution, taken on May 13, 2013).

The color and shape characteristics of landslides (e.g., morphology, hue, shadow, texture) of the optical remote sensing

images is clearly distinguishable from the surrounding area, especially shortly after the earthquake. Following the criteria



proposed by Harp et al. (2011) and Xu (2015), we manually mapped the earthquake-triggered landslides using the GIS platform.

Coseismic landslides are easily detected on high-resolutions images (~1–10 m or higher). Many coseismic landslides may overlap in areas of high landslide density, which makes them difficult to uniquely distinguish and subject to the interpreter's discretion. Densely vegetated areas also pose a problem because vegetation is often destroyed during landsliding and "tadpole" traces appear on the image. Special care must be taken to distinguish the landslide from the vegetation damage range and the identified landslide inventory map must be validated to improve the interpretation accuracy. To address this problem, we first removed landslides in areas with slopes less than 20° because landslide events require certain terrain conditions. Because some farmland, bare land, quarries, sand and gravel yards, old landslides, and other traces of human activities are difficult to distinguish from coseismic landslides on the images, we performed detailed field investigations and combined pre-earthquake satellite images to exclude inaccurate assessments. Finally, we converted the projected landslide area identified in the remote sensing image with the slope gradient to obtain the final landslide area.

## 2.3 Analysis and results

Although the distribution of earthquake-induced landslides is controlled by a variety of factors (e.g., slope aspect, faulting, topography, lithology) that have been discussed in several studies, we believe that the most important control factor is seismic intensity. Therefore, in this study, we analyze landslide distribution according to seismic intensity zone. The range of seismic intensity zones is based on official seismic intensity distribution maps published by the China Earthquake Administration.

Frequency-area (or volume) distribution of a landslide event often exhibits power-law scaling over a limited scale range(Guzzetti et al. 2002; Stark 2001).  The relationship can be represented by Eq. (1):

$$N(A) = a \times A - b ,$$ (1)

where a , b is constants, b is the exponent power that describes the statistical features of the landslide distribution. $A$ is landslide size,  characterized by area and volume, even depth. $N(A)$ is the number of landslides beyond a given $A$.

### 2.3.1 Landslide volume (depth)-frequency distributions in different seismic intensity zones

We performed detailed field investigations of $10^5$ landslides triggered by the Wenchuan earthquake and 261 landslides triggered by the Lushan earthquake. The data were fitted by Eq. (1) where $q$ is landslide volume and $N(q)$ is the number of landslides beyond a given volume $q$, $h$ is landslide depth and $N(h)$ is the number of landslides beyond a given depth $h$. The power-law formula is fitted by a least-squares regression where $R^2$ represents the goodness of fit (i.e., larger $R^2$ reflect a better fit). The results are listed in Table 1.





**Table 1: Landslide volume (depth)-frequency distributions in different seismic intensity zones**

| seismic intensity | Landslides number | volume -frequency fitting | (depth)-frequency fitting |
|---|---|---|---|
| Lushan earthquake | | | |
| VII | 108 | $N(q) = 485.69q^{-0.529}$, $R^2$=0.9647 | - |
| VIII | 108 | $N(q) = 413.97q^{-0.518}$, $R^2$=0.9864 | - |
| IX | 45 | $N(q) = 214.58q^{-0.611}$, $R^2$=0.9465 | - |
| Wenchuan earthquake | | | |
| IX | 61 | $N(q) = 223.14q^{-0.483}$, $R^2$=0.9639 | $N(h) = 22.38q^{-1.037}$, $R^2$=0.9871 |
| X | 29 | $N(q) = 260.76q^{-0.565}$, $R^2$=0.8940 | $N(h) = 11.18q^{-1.243}$, $R^2$=0.8963 |

In the VII, VIII, and IX seismic intensity zones of the Lushan earthquake, the landslides volume-frequency distribution follows a strong power-law relationship. In the IX seismic intensity zone of the Wenchuan earthquake, the volume-frequency distribution and depth-frequency distribution all follow a power-law relationship. The X seismic intensity zone of the Wenchuan earthquake shows similar yet weak power-law characteristics, even though the number of landslides is small (29

sites). A substantial number of large landslides are observed in the XI seismic intensity zone of the Wenchuan earthquake and are connected in a single mass, which is not easy to distinguish into individual sites. Almost two-thirds of the mountains show evidence of mountain peeling. Of the 15 landslides surveyed, the minimum volume is 20 m$^3$ and the maximum is 16800 m$^3$. The minimum depth is 0.3 m and the maximum is 5 m. Although the number of samples is insufficient to draw statistical conclusions, large landslide events appear to dominate. A power-law relationship is not observed between the

volume (or depth) and frequency in the XI seismic intensity zone.

### 2.3.2 Landslide frequency-area distributions in different seismic intensity zones

We identified 20,254 coseismic landslides of the Wenchuan earthquake and 1608 coseismic landslides of Lushan earthquake by manual image interpretation, as shown in Figs. 2 and 3, respectively. The statistics of the landslide area were fitted by Eq. (1), where $A$ is landslide area and $N(A)$ is the number of landslides beyond a given area $A$. To eliminate the influence of map

resolution and undersampling of smaller landslides, we statistically analyze landslides over 1000 m$^2$. The power-law formula is fitted by a least-squares regression. We also test the lognormal relationship of the data by the chi-square test method. The results are listed in Table 2.



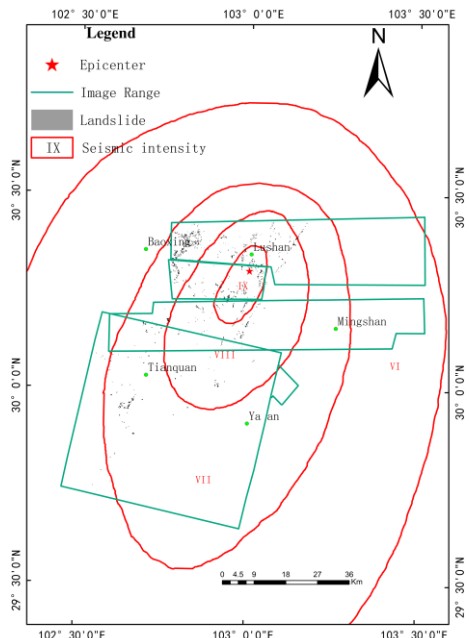

**Figure 2: Distribution map of landslides triggered by the Wenchuan earthquake**

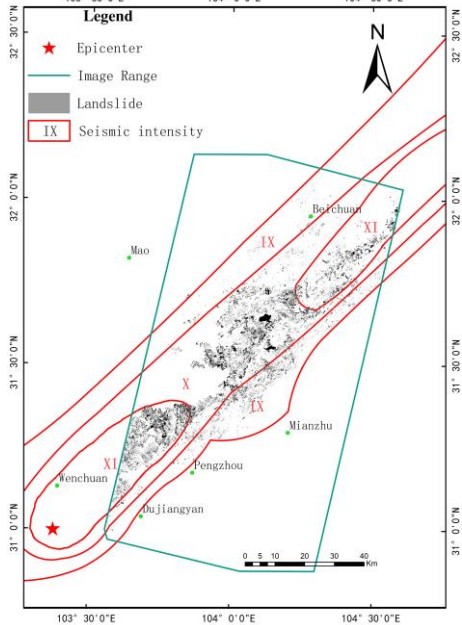


**Figure 3: Distribution map of landslides triggered by the Lushan earthquake**





**Table 2: Landslide frequency-area distributions in different seismic intensity zones**

| Seismic intensity | Number of landslides | Fitting formula | Hypothesis test result |
|---|---|---|---|
| | | Lushan earthquake | |
| VII | 706 | $N(A) = 5.5 \times 10^5 A^{-1.051}$ | $R^2$=0.901, Accept the power law distribution hypothesis |
| VIII | 477 | $N(A) = 1.6 \times 10^5 A^{-0.993}$ | $R^2$=0.917, Accept the power law distribution hypothesis |
| IX | 425 | $N(A) = 6.8 \times 10^4 A^{-0.955}$ | $R^2$=0.916, Accept the power law distribution hypothesis |
| | | Wenchuan earthquake | |
| IX | 3775 | $N(A) = 1.7 \times 10^9 A^{-1.543}$ | $R^2$=0.913, Accept the power law distribution hypothesis |
| X | 9615 | $N(A) = 1.02 \times 10^9 A^{-1.304}$ | $R^2$=0.873, Accept the power law distribution hypothesis |
| XI | 6846 | $N(A) = \dfrac{1}{1.46\sqrt{2\pi A}} e^{\frac{-(LnA-9.02)^2}{2\times 1.46^2}}$ | Accept the lognormal distribution hypothesis at the significance level of 0.05 |

In the VII, VIII, and IX seismic intensity zones of the Lushan earthquake, the landslides area-frequency distribution follows a power-law relationship ($R^2 > 0.9$), although none pass the lognormal distribution test. The landslides area-frequency distribution follows a power-law relationship ($R^2 > 0.9$) in the IX seismic intensity zone of the Wenchuan earthquake, a weak power law in the X seismic intensity zone ($R^2$=0.873), while a lognormal distribution in the XI seismic intensity zone.

### 2.3.3 Landslide density in different seismic intensity zones

We define landslide density as the number of landslides per square kilometer. The landslide numbers and calculated densities in the different seismic intensity zones of the two earthquakes are listed in Table 3.

**Table 3: Landslide concentrations in different seismic intensity zones**

| Seismic intensity | Number of landslides | Total area (km²) | Landslide concentration (landslides per km²) |
|---|---|---|---|
| Lushan earthquake | | | |
| VII | 706 | 2068 | 0.34 |
| VIII | 477 | 1161 | 0.41 |





| IX | 425 | 189 | 2.25 |
|---|---|---|---|
| Wenchuan earthquake | | | |
| IX | 3775 | 1645 | 2.30 |
| X | 9615 | 1627 | 5.91 |
| XI | 6846 | 1074 | 6.37 |

Landslide density clearly increases with seismic intensity, in agreement with previous studies (Qi et al., 2010).To test if the statistical results obtained from landslides generated by the Wenchuan and Lushan earthquakes may also apply to landslides in other areas, we combine cellular automaton simulations with laboratory experiments under the conceptual framework of SOC to interpret the distribution mechanism of earthquake-triggered landslides from a physical viewpoint.

## 3 Cellular automata simulations

Computer models are an integral component of SOC research, mostly from numerical simulations of a sand pile cellular automaton. The Bak, Tang, and Wiesenfeld (1987) model (BTW) is the earliest and most classic model of sand pile cellular automata and many other models have been developed based on the BTW model for different physical systems that display SOC (e.g., earthquakes, forest fires, magnetic vortex motion, interface growth, biological evolution). In this study, we introduce a sand pile cellular automata model to simulate earthquake-induced landslides.

### 3.1 Model procedure

According to the physical features of earthquake-induced landslides, the model has following aspects:

(1) The BTW model is used to study the avalanche size distribution over time of a sand pile under disturbances. Because earthquake-induced landslides occur simultaneously, the avalanche size distribution of many sand piles must be studied when they are disturbed by a single event.

(2) In the BTW model, a system may be partially disturbed and remain strictly energetically conservative. During earthquake-induced landslides, the slope is disturbed as a whole and its instability must overcome its self-stabilization ability, which requires input energy. The disturbed disseminate is therefore energetically non-conservative.

(3) The BTW model is used to study the dynamic behavior of a system under the same perturbation conditions. For earthquake-induced landslides, the disturbance force on the slope is sensitive to the seismic intensity zone and may exceed the perturbation level. We must therefore modify the disturbance intensity to simulate this physical process.

According to the above characteristics, we constructed a sand pile cellular automata model to simulate earthquake-induced landslides. The model is defined on a two-dimensional $L \times L$ lattice. The sites were numbered with a pair of sub-indexes





(i, j)(1 ≤ i, j ≤ L), and each site has four nearest neighbors located in the upper, lower, left, and right directions. The state

of each site is characterized by a non-negative integer variable $F_{i,j}$, which is the state value that reflects the stability of site

(i, j) (equivalent to the site energy) and each site has a threshold $F_{th}$. We introduce a as the disturbed transmission parameter

for the four neighbors, which are not larger than 0.25 owing to non-conservative factors. The model is described by the

following algorithm.

Step 1. N sand piles of equal size are built simultaneously but with different initial states. For each sand pile, all sites are

initialized to a random value between 0 and $F_{th}$, and Fmax is the maximum value of all sites in the sand pile.

Step 2. The N sand piles are simultaneously disturbed. The state value of all sites increases uniformly by the disturbance

intensity $F'$.

$$F_{i,j} \rightarrow F_{i,j} + F' \tag{2}$$

Step 3. The N sand piles are investigated individually. If all sites in a given sand pile remain less than $F_{th}$, nothing happens.

Conversely, if $F_{i,j} \geq F_{th}$, the site (i, j) becomes unstable and relaxes according to the rule:

$$F_{i\pm1,j} \rightarrow F_{i\pm1,j} + \alpha F_{i,j}$$

$$F_{i,j\pm1} \rightarrow F_{i,j\pm1} + \alpha F_{i,j}$$

$$F_{i,j} \rightarrow 0 \tag{3}$$

The relaxation may cause some of the neighbors to become unstable. If so, step 3 is repeated all sites are less than $F_{th}$

$(F_{i,j} < F_{th})$.

By changing $F'$, we can determine the relationship between avalanche magnitude and occurrence frequency.

It should be noted that when a disturbance is applied, some sand piles will react while others will not depending on $F'$. When

$F' = F_{th} - F_{max}$, at least one sand pile will react. When $F'$ is small, sometimes only a few sites will be triggered. But as $F'$

increases, a batch of sites may be triggered, each of which may trigger chain reactions that may ultimately cross paths in

space. Parallel processing is therefore adopted in the algorithm and all disturbed sites react simultaneously in a parallel-

updating manner. We measured avalanche size in terms of the number of sites participating in the relaxation. This property is

called cluster size and is as a measure of the area affected by the avalanche.

### 3.2 Results

The model parameters are $\alpha = 0.2$, $F_{th} = 1$, and $L = 50$. One million sand piles (N = $10^6$) were generated. Each sand pile

was continuously reacted $10^5$ times with a disturbance intensity of $F' = 1 - F_{max}$ in succession to ensure that each sand pile

evolves to a critical state before the formal experiment. Eight groups of simulation experiments were then carried out by

increasing $F'$ from 0.00001 to 0.01, with each group reacted only once. Let the number of avalanches be $S$ and the frequency





of the avalanche size equal to $S$ be $f(S)$. The avalanche density is equal to the number of sand piles with avalanche events divided by the total number of sand piles. The statistical results are shown in Table 4.

**Table 4: Results of cellular automata simulations**

| No. | disturbance intensity $F'$ | Number of sandpiles with avalanche event | Sandpile numbers | avalanche density ($\rho$) | Fitting formula | Hypothesis test result |
|---|---|---|---|---|---|---|
| 1 | 0.000 01 | 80049 | $10^6$ | 0. 08 | $f(S) = 0.229S^{-2.51}$ | $R^2$=0.949, Accept the power law distribution hypothesis |
| 2 | 0.000 05 | 286511 | $10^6$ | 0.29 | $f(S) = 0.576S^{-2.44}$ | $R^2$=0.964, Accept the power law distribution hypothesis |
| 3 | 0.000 1 | 468255 | $10^6$ | 0.47 | $f(S) = 1.168S^{-2.46}$ | $R^2$=0.965, Accept the power law distribution hypothesis |
| 4 | 0.000 5 | 936746 | $10^6$ | 0.94 | $f(S) = 1.327S^{-2.56}$ | $R^2$=0.964, Accept the power law distribution hypothesis |
| 5 | $1 - F_{max}$ | $10^6$ | $10^6$ | 1 | $f(S) = 1.267S^{-2.01}$ | $R^2$=0.969, Accept the power law distribution hypothesis |
| 6 | 0.001 | $10^6$ | $10^6$ | 1 | $f(S) = 117.41S^{-2.94}$ | $R^2$=0.901, Accept the power law distribution hypothesis |
| 7 | 0.005 | $10^6$ | $10^6$ | 1 | $f(S) = \dfrac{1}{0.54\sqrt{2\pi}S} e^{\frac{-(LnS-3.88)^2}{2\times0.54^2}}$ | Accept the lognormal distribution hypothesis at the significance level of 0.05 |
| 8 | 0.010 | $10^6$ | $10^6$ | 1 | $f(S) = \dfrac{1}{0.38\sqrt{2\pi}S} e^{\frac{-(LnS-4.68)^2}{2\times0.38^2}}$ | Accept the lognormal distribution hypothesis at the significance level of 0.05 |

Table 4 shows that the disturbance intensity $1 - F_{max}$ can be divided into two intervals and the dynamic characteristics of the sand pile model exhibit different properties. When $F' < (1 - F_{max})$, the avalanche scale and occurrence frequency



basically obey the same power-law distribution but the avalanche density increases monotonously with increasing $F'$ (Fig. 4).

When $F' > (1 - F_{max})$, the dynamics of the sand pile model exhibit a strong to weak power-law relationship and then to a

lognormal distribution with increasing $F'$ (Fig. 5).

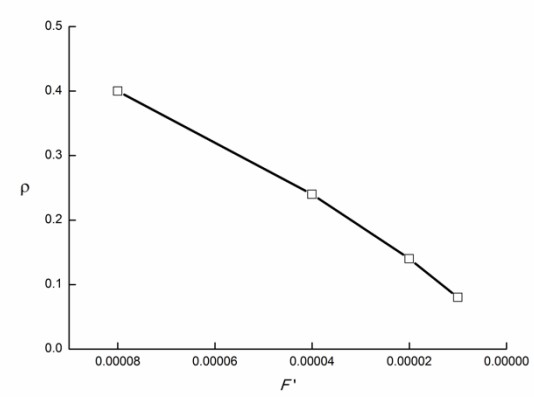

**Figure 4: Avalanche density $\rho$ vs. disturbance intensity $F'$**

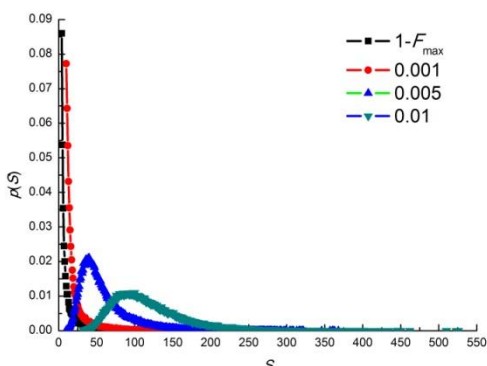

**Figure 5: Number of events corresponding to the cluster size distribution**

**4 Shaking table sand pile model tests**

Although cellular automata numerical simulations are the primary way to obtain properties of a SOC system, physical experiments are necessary to verify the validity of its application to earthquake-trigged landslides. A sand pile is a classic example of SOC. Held et al. (1990) designed a physical experiment to show that a sand pile is indeed a SOC system and subsequent studies have carried out on various types of sand pile experiments to determine the mechanism of certain





physical systems that show SOC. To better understand the physical phenomena of earthquake-induced landslides, we performed shaking table tests to study the dynamic behavior of a sand pile under different earthquake forces.

## 4.1 Experimental procedure

A landslide triggered by an earthquake is a natural phenomenon that occurs over a tremendously large size range ($\sim10^2 - 10^8$ m$^3$). The purpose of the experiment is to study the dynamic behavior of a sand pile and the model sand piles need not

simulate a certain prototype. Previous sand pile experiments have shown that the gradation of model material, physical and mechanical parameters, and model size may influence the collapse size, but there is no influence on the relationship between collapse size and its occurrence frequency. We therefore did not consider similarity relations in the tests.

Large-scale shaking table experiments were conducted in the Key Laboratory of High-speed Railway Engineering at Southwest Jiaotong University in China (Fig. 6). The shaking table is a single-direction table with a size of $2 \times 4$ m, capacity

of 25,000 kg, and loading frequency range of 0.4–15 Hz. In the absence of an applied load, the maximum acceleration is 1.2 g and the displacement ranges from −100 to 100 mm.

The one-side slope sand pile was built in a steel model box with a 3.75-m length, 1.75-m width, and 2.75-m height placed on the shaking table. The sand pile material was a dried natural sand gravel collected from earthquake-triggered landslides in the Longmen Mountains. Particles larger than 50 mm were removed and the gradation was measured (Fig. 7). When the sand

pile reaches its natural angle (e.g., soil angle of internal friction), it is in a critical stable state. The sand pile has a length of 2.58 m, width of 1.5 m, height of 1.95 m, and total weight of 6800 kg (Fig. 8).

Slope responses under the excitation of field seismic waves were recorded at the Wolong seismic station of the 2008 Wenchuan earthquake (WL wave, Fig. 9). The input WL wave was proportionally scaled to its peak value.

To study the variation of sand pile dynamic characteristics with increasing disturbance intensity, we designed five sets of

tests with input peak accelerations of 0.075 g to 0.250 g. After inputting the excitation, the weight of the sand gravel collapse was measured as a test value. After each test, sand gravel was added from the top of the slope to ensure that the slope remained in a critical stable state. Each set of tests was run no less than 60 times.




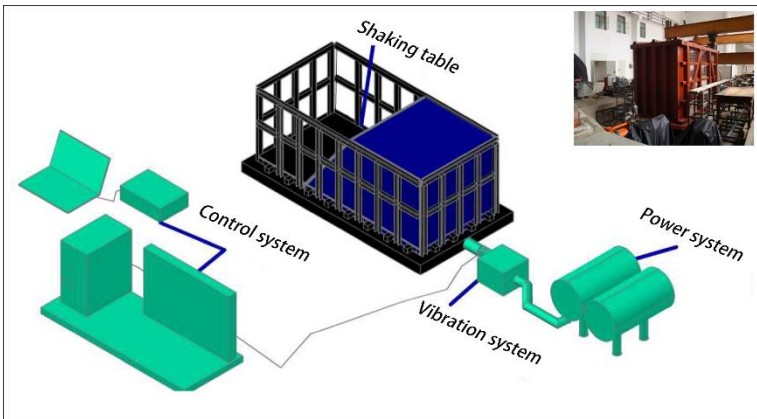

**Figure 6: Schematic diagram of the shaking table experimental setup**

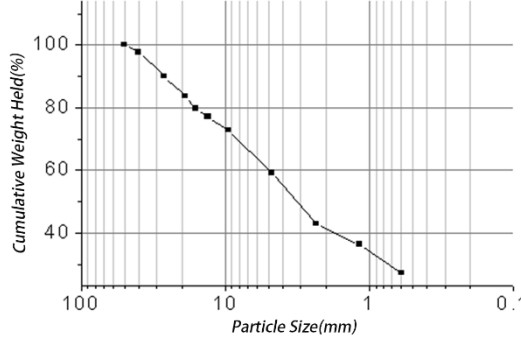


**Figure 7: Gradation curve of the sand sample**





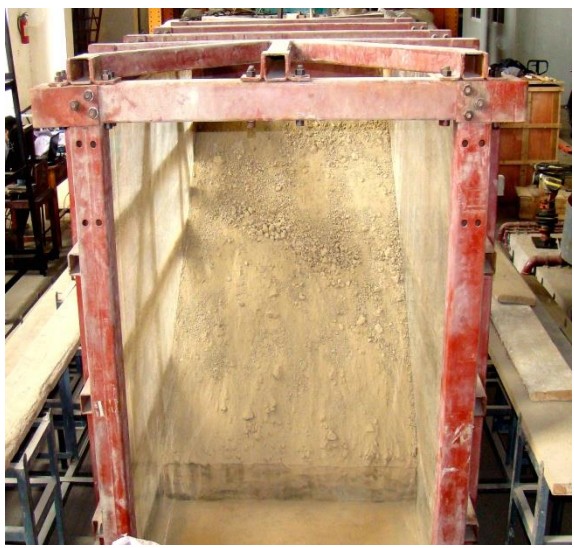

**Figure 8: Photo of the sand pile model**

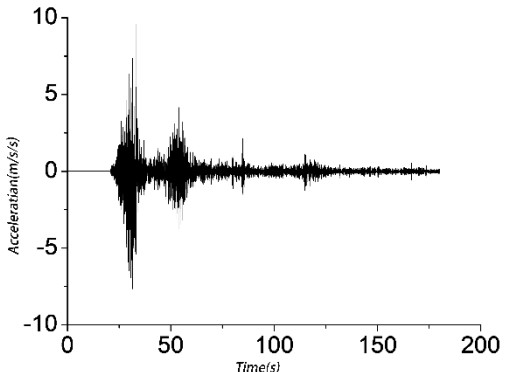

Figure 9: Acceleration history of the WL wave with a peak value of 976 gal

**4.2 Data analysis of result**

Let the collapse weight be $x$ and the collapse weight frequency equal to be $f(x)$. The analysis results are shown in Table 5.

The collapse density is equal to the number of tests with collapse events divided by the total number of tests.

**Table 5: Statistical results of sand pile tests**





| No. | Peak acceleration | Number of tests | Number of tests with collapse event | collapse density | Fitting formula | Hypothesis test result |
|---|---|---|---|---|---|---|
| 1 | 0.075g | 90 | 49 | 0.54 | $f(x) = 500.2x^{-0.774}$ | $R^2$=0.901, accept the power law distribution hypothesis |
| 2 | 0.100 g | 90 | 52 | 0.58 | $f(x) = 579.3x^{-0.783}$ | $R^2$=0.917, accept the power law distribution hypothesis |
| 3 | 0.125g | 150 | 118 | 0.79 | $f(x) = 3887.6x^{-1.059}$ | $R^2$=0.963, accept the power law distribution hypothesis |
| 4 | 0.150g | 60 | 60 | 1 | $f(x) = \dfrac{1}{0.59\sqrt{2\pi}x} e^{\frac{-(Lnx-7.36)^2}{2\times0.59^2}}$ | Accept the lognormal distribution hypothesis at the significance level of 0.05 |
| 5 | 0.250g | 60 | 60 | 1 | $f(x) = \dfrac{1}{0.32\sqrt{2\pi}x} e^{\frac{-(Lnx-8.33)^2}{2\times0.32^2}}$ | Accept the lognormal distribution hypothesis at the significance level of 0.05 |

Note: When the PGA is 0.075 g, 0.1 g, and 0.125 g, some tests occurred without collapse events and the number of tests was increased. The other groups were repeated 60 times.

When the peak acceleration input was between 0.075 g and 0.125 g, some tests occurred without collapses. When collapses did occur, small collapses were significantly more common than large collapses and the results obey a power-law distribution. When the PGA was between 0.15 g and 0.25 g, no tests occurred with zero collapses, the power-law relationship was weakened, and the results followed a lognormal distribution. The collapse density increases with increasing peak acceleration.

Studying the dynamic response of sand pile with seismic waves as disturbance sources is a unique experimental method to study SOC characteristics. Previous results have shown that the disturbance mode does not affect the sand pile dynamics characteristics, but does influence the power-law relationship parameters (Yao et al.,1998, 2003; Yang et al., 2007). The shaking table sand pile model tests show that changes in disturbance intensity lead to a shift in system dynamics. The



physical process of the shaking table sand pile test is close to the prototype problem of earthquake-induced landslides, even though the number of experiments remains limited, and provides good support for the universality of earthquake-triggered landslides in different intensity zones.

**6 Conclusions and discussion**

(1) We analyzed data from landslides triggered by the 2013 Lushan and 2008 Wenchuan earthquakes. The results show a negative power-law relationship between landslide size and frequency in the VII, VIII, and IX seismic intensity zones. The relationship becomes a weak power law in the X seismic intensity zone and changes into a lognormal distribution in the XI seismic intensity zone. Landslide density increases gradually with increasing seismic intensity. Cellular automaton

simulations reveal that with increasing disturbance intensity, the dynamical mechanism of the sand pile model changes from a strong power law to a weak power law and then to a lognormal distribution, and the avalanche density increases. The results of the shaking table sand pile model tests verify these findings. The overall landslide distribution law is therefore constrained, even though these landslides are complex and very random, and the evolution mechanism of the distribution law in different intensity zones is clarified. The distribution probability model of earthquake-triggered landslides and evolution

model with increasing seismic intensity presented here exceed the statistical relation level of a typical sample, which is therefore a universal law.

(2) SOC was founded in 1987 as a branch of non-equilibrium thermodynamics and study of its phenomenology and precise definition continues. SOC has highlighted that thresholds, metastability, and large-scale fluctuations play a decisive role in the spatiotemporal behavior of a large class of multi-body systems. However, the influence of disturbance intensity on the

system dynamics behavior of SOC has not received much attention. The disturbance intensity range of a catastrophic event may span several orders of magnitude (e.g. the energy difference between a magnitude 9 earthquake is 32,768 times more powerful than a magnitude 6 earthquake). In the case of the 2008 Wenchuan and 2013 Lushan earthquakes, the dynamic characteristics of the SOC system undergo a strong power law, weak power law, and lognormal distribution evolution process with increasing disturbance intensity. This constraint on the evolution pattern of a SOC system behavior goes

beyond the traditional field of SOC and makes a strong impetus to further develop basic SOC theory.

(3) Compared with chaos theory, SOC does not greatly emphasize the initial conditions and system details, which facilitates the experimental and analytical procedures. The sand pile example tends to explain nearly everything from mountain formation to stock market volatility. But if many of the unique details of mountain systems can be understood by simple cellular automata numerical simulations, it may not be realistic for most geographers and requires further validation. For

example, slope structures (e.g., joints, fracture surfaces) in nature are non-uniformly distributed and natural granular materials often exhibit a wide gradation. The nonuniformity of components is one of the important characteristics of a





mountain system, however, the influence of non-uniform components on the dynamics is not considered here. We briefly discuss the effect of nonuniformity on dynamics in a previous study (Guo et al., 2017). However, the heterogeneity of cell geometry, arrangement randomness, and interaction anisotropy can be considered to investigate the unique details of how

nonuniformity property affects system dynamics. Further research will therefore aim to determine the deep-seated law of earthquake-induced landslides.

 (4) In the IX seismic intensity zone, the cumulative number-area distribution of landslides triggered by the earthquakes exhibits a negative power-law relationship but with different power exponents: 1.543 for the Wenchuan earthquake and 0.955 for the Lushan earthquake. Whether the observed variability in the power-law exponents is incidental or applies only

under certain conditions should be investigated in future studies.

**Acknowledgments**

This work was supported by the National Natural Science Foundation of China (Grant No. 41902302), the International S&T Cooperation Project of the Chinese Academy of Sciences (Grant No.131551KYSB20180042), Sichuan Science and Technology Program of China (Grant Nos. 20GJHZ0205, 2019YFG001) and the Project of China Railway Eryuan

Engineering Group Co. Ltd. (Grant No. KYY2020054(20-22)). We would like to express our gratitude to Dr. Chenwen Guo and Dr. Haiqiang Guo for their helpful contributions to this article. We thank Esther Posner, PhD, from Liwen Bianji, Edanz Editing China (www.liwenbianji.cn/ac), for editing the English text of a draft of this manuscript.

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
