# Peer review of "Size distribution law of earthquake-triggered landslides in different seismic intensity zones"

_Nonlinear Processes in Geophysics, 2020_

## Referee Comment (RC1) · Anonymous Referee #1 · 24 Nov 2020

The scale frequency distribution of earthquake-triggered landslides is an important issue. This manuscript analyzes the scale frequency relationship of landslides based on the spatial distribution of landslides and physical experiments. Through the comparison and comprehensive analysis of the revealed laws of spatial distribution and the phenomena of physical experiments, some useful conclusions are addressed. The special comments include: (1) The author considered that the scale frequency relationship of landslides developed in intensity VII-IX conforms to one law, that of landslides in intensity x conforms to another law, and that in intensity XI conforms to another law. First of all, there are very few landslides in VII and VIII, and there will be a great deal of uncertainty in the law obtained from such a small sample. (2) The ground motion

accelerations used in the sand pile experiments carried out by the authors are 0.075-0.125g and 0.15-0.25g, although the phenomena claimed by the author are consistent with those obtained from the spatial distribution of landslides. However, 0.075-0.125g does not correspond to VII-IX intensity area. In fact, in general, 0.2g corresponds to the VIII degree region. Obviously, the boundary conditions of spatial analysis are completely different from those obtained by physical experiments, and it seems that there is no comparability. (3) The landslide data of different intensities used in this paper come from two earthquake events. Wenchuan earthquake is mainly high intensity area, while Lushan earthquake is mainly medium intensity area. Undoubtedly, the difference of different seismic characteristics and the nature of earthquake affected area, such as geological and topographical conditions, may greatly affect the results, so that they are not comparable. (4) The data quality of landslides triggered by the two earthquakes does not seem to be perfect. Compared with the existing work, either the whole earthquake area is not covered, or many landslides seem to be missed. There is no doubt that the law revealed by such data may deviate from the actual situation. In conclusion, a major revision is recommended.

---

## Referee Comment (RC2) · Anonymous Referee #2 · 28 Nov 2020

General comments The manuscript shows the result of size distribution law of earthquake-triggered landslides in different seismic intensity zones. The research is useful for regional scale landslide hazard and risk assessment. However, much effort should be done to clarify or deepen the obtained results.

Special comments 1.From the title and the abstract, I thought the objective of the paper was to find the distribution law between size and frequency of landslides triggered by earthquake with different intensities. However, the paper structure should be well managed considering three ways (inventory data, computer simulation and physical experiment) in the paper. 2. Landslide inventory data in the paper is not clearly clarified.

The process and existing uncertainties in image interpretation for landslides should be explained. How did the authors deal with the connected landslides which are common in Wenchuan earthquake events and difficult to be separated? The author states the phenomena in lines 160-161, but without any other words later. 3. The distribution law of volume (depth)-frequency and area-frequency is obtained and shown in Table 1 and Table 2. The results are from the same triggering events but the number of the samples shown in the tables is not the same. Why? 4. Equation 1 is wrong. 5. Please explain the matching ability of the physical experiment with the real earthquake events, such as in terms of the peak acceleration in experiment and seismic intensity in Table 2-3.

Technical corrections 1. Grammar mistake exists in the paper, such as Lines 30-34, Line 211. 2. The quality of the figures need to be improved. 3. The unit of the parameters in Line 234 and Figure 4-5 is not clear. 4. The language needs to be improved.

---

## Author Comment (AC1) · 11 Jan 2021

We very much appreciate the careful reading of our manuscript and valuable suggestions of the reviewer.Overall the comments have been fair, encouraging and constructive. We have learned much from it. Responds to the reviewer's comments: Comment 1: The author considered that the scale frequency relationship of landslides developed in intensity VII-IX conforms to one law, that of landslides in intensity x conforms to another law, and that in intensity XI onforms to another law. First of all, there are very few landslides in VII and VIII, and there will be a great deal of uncertainty in the law obtained from such a small sample. Response: Landslide data were accessed through

by field investigation and interpretation of high-resolution remote sensing images. We got 108 landslides by field investigation and 706 by remote sensing interpretation in VII zone, 108 landslides by field investigation and 477 by remote sensing interpretation in VIII zone. The field survey data is small because we can only measure the landslides visible along the highway. Furthermore, this study divides the landslide data into seismic intensity zones for statistics, so the samples in each zone are not large. In the study of frequency-magnitude distribution of landslides, the sample number of different landslides databases are quite different. For example, Brunetti (2009) examined 19 landslide datasets. Individual datasets include from 17 to 1019 landslides of different types. Each landslide dataset exhibits heavy tailed (self-similar) behaviour for their frequency-size distributions. (References:Brunetti M., Guzzetti F., and Rossi M.: Probability distributions of landslide volumes. Nonlinear Processes in Geophysics, 16(2):179-188. https://doi.org/10.5194/npg-16-179-2009,2009.)

Comment 2: The ground motion accelerations used in the sand pile experiments carried out by the authors are 0.075-0.125g and 0.15-0.25g, although the phenomena claimed by the author are consistent with those obtained from the spatial distribution of landslides. However, 0.075-0.125g does not correspond to VII-IX intensity area. In fact, in general, 0.2g corresponds to the VIII degree region. Obviously, the boundary conditions of spatial analysis are completely different from those obtained by physical experiments, and it seems that there is no comparability. Response: In the shaking table experiments, the one-side slope sand pile was built by dried sand gravel reaching its natural angle. The strength of the loose slope is less than the natural slope. In

---

## Author Comment (AC2) · 11 Jan 2021

We very much appreciate the careful reading of our manuscript and valuable suggestions of the reviewer. Responds to the reviewer's comments: Comment 1: From the title and the abstract, I thought the objective of the paper was to find the distribution law between size and frequency of landslides triggered by earthquake with different intensities. However, the paper structure should be well managed considering three ways (inventory data, computer simulation and physical experiment) in the paper. Response: Thanks for your comments, I'll try to revise to make it more focused.

Comment 2: Landslide inventory data in the paper is not clearly clarified. The pro-

cess and existing uncertainties in image interpretation for landslides should be explained. How did the authors deal with the connected landslides which are common in Wenchuan earthquake events and difficult to be separated? The author states the phenomena in lines 160-161, but without any other words later. Response: Several individual landslides in low-resolution images may be misjudged into a single larger landslide because the distances of them are less than the resolution of the image. Therefore, the higher resolution of the images used, the easier individual landslides are separated and the more objective landslide inventory maps prepared. The remote sensing data collected in this paper have high resolution, and the single landslide can be identified through manual visual inspection combined with experience.

Comment 3: The distribution law of volume (depth)-frequency and area-frequency is obtained and shown in Table 1 and Table 2. The results are from the same triggering events but the number of the samples shown in the tables is not the same. Why? Response: The samples in Table 1 are obtained from field surveys, and the samples in Table 2 are from remote sensing interpretation. The number of samples obtained from field surveys is small, while that from remote sensing interpretation is large.

Comment 4: Equation 1 is wrong. Response: I am very sorry for our incorrect writing, I have revised it. The correct equation is: $N(A)=a\times A^{(-b)}$

Comment 5: Please explain the matching ability of the physical experiment with the real earthquake events, such as in terms of the peak acceleration in experiment and seismic intensity in Table 2-3. Response: In general, 0.1g corresponds to the VII degree region, 0.2g corresponds to the VIII degree region, 0.4g corresponds to the IX degree region.In the physical experiment, 0.075-0.125g corresponds to the region blow IX degree, and 0.15-0.25g corresponds to the region above IX degree. In the shaking table experiments, the one-side slope sand pile was built by dried sand gravel reaching its natural angle. The strength of the loose slope is less than the natural slope. In other words, a small peak acceleration could cause the sand pile to collapse in the experiment. Therefore, the seismic intensity in the physical experiment is less

than in the real earthquake events. The purpose of the experiment is to study the evolution trend of the distribution law with increasing seismic intensity, which is mainly a comparison of the evolution law without a definite numerical correspondence.

Technical corrections 1. Grammar mistake exists in the paper, such as Lines 30-34, Line 211. 2. The quality of the figures need to be improved. 3. The unit of the parameters in Line 234 and Figure 4-5 is not clear. 4. The language needs to be improved. Response: Thank you for pointing out the mistakes in our writing, we have corrected.

---

## Author Comment (AC3) · 11 Jan 2021

We very much appreciate the careful reading of our manuscript and valuable suggestions of the reviewer.Overall the comments have been fair, encouraging and constructive. We have learned much from it. Responds to the reviewer's comments: Comment 1: The author considered that the scale frequency relationship of landslides developed in intensity VII-IX conforms to one law, that of landslides in intensity x conforms to another law, and that in intensity XI onforms to another law. First of all, there are very few landslides in VII and VIII, and there will be a great deal of uncertainty in the law obtained from such a small sample. Response: Landslide data were accessed through

by field investigation and interpretation of high-resolution remote sensing images. We got 108 landslides by field investigation and 706 by remote sensing interpretation in VII zone, 108 landslides by field investigation and 477 by remote sensing interpretation in VIII zone. The field survey data is small because we can only measure the landslides visible along the highway. Furthermore, this study divides the landslide data into seismic intensity zones for statistics, so the samples in each zone are not large. In the study of frequency-magnitude distribution of landslides, the sample number of different landslides databases are quite different. For example, Brunetti (2009) examined 19 landslide datasets. Individual datasets include from 17 to 1019 landslides of different types. Each landslide dataset exhibits heavy tailed (self-similar) behaviour for their frequency-size distributions. (References:Brunetti M., Guzzetti F., and Rossi M.: Probability distributions of landslide volumes. Nonlinear Processes in Geophysics, 16(2):179-188. https://doi.org/10.5194/npg-16-179-2009,2009.)

Comment 2: The ground motion accelerations used in the sand pile experiments carried out by the authors are 0.075-0.125g and 0.15-0.25g, although the phenomena claimed by the author are consistent with those obtained from the spatial distribution of landslides. However, 0.075-0.125g does not correspond to VII-IX intensity area. In fact, in general, 0.2g corresponds to the VIII degree region. Obviously, the boundary conditions of spatial analysis are completely different from those obtained by physical experiments, and it seems that there is no comparability. Response: In the shaking table experiments, the one-side slope sand pile was built by dried sand gravel reaching its natural angle. The strength of the loose slope is less than the natural slope. In other words, a small peak acceleration could cause the sand pile to collapse in the experiment. Therefore, the seismic intensity in the physical experiment is less than in the real earthquake events. The purpose of the experiment is to study the evolution trend of the distribution law with increasing seismic intensity, which is mainly a comparison of the evolution law without a definite numerical correspondence.

Comment 3: The landslide data of different intensities used in this paper come from two

earthquake events. Wenchuan earthquake is mainly high intensity area, while Lushan earthquake is mainly medium intensity area. Undoubtedly, the difference of different seismic characteristics and the nature of earthquake affected area, such as geological and topographical conditions, may greatly affect the results, so that they are not comparable. Response: In general, landslides triggered by earthquakes are rare events, and the probability of two earthquakes occurring in the same area is even lower, providing valuable information for scientific research. Both the Wenchuan and Lushan earthquakes occurred within the Longmen Mountain fold-and-thrust belt. There is a gap extending about 75km between the main-shock of 2008 Wenchuan and Lushan events(Fig.1). Most scholars believe that the terrain and geological conditions are similar, and the thrust mechanism is similar. Some even suggest that Lushan earthquake was a strong aftershock after Wenchuan earthquake. In the study of landslide distribution, the distribution of landslides triggered by different evens (e.g., earthquake, rainfall) is often compared (Dussauge et al., 2003; Brunetti et al., 2009). In this paper, the landslide distribution is analyzed according to different seismic intensity, which is different from other similar studies. (References: Dussauge, C., Grasso, J. R., and Helmstetter, A. Statistical analysis of rockfall volume distributions: implications for rockfall dynamics, J. Geophys. Res., 108(B6), 2286, doi:10.1029/2001JB000650, 2003. Brunetti M., Guzzetti F., and Rossi M.: Probability distributions of landslide volumes. Nonlinear Processes in Geophysics, 16(2):179-188. https://doi.org/10.5194/npg-16-179-2009,2009.)

Comment 4: The data quality of landslides triggered by the two earthquakes does not seem to be perfect. Compared with the existing work, either the whole earthquake area is not covered, or many landslides seem to be missed. There is no doubt that the law revealed by such data may deviate from the actual situation. Response: Because of the difference of image precision, interpretation method and evaluation standard, the number of landslides triggered by Lushan earthquake and Wenchuan earthquake obtained by different experts is quite different. Typical examples are: For Wenchuan earthquake, Huang Runqiu et al. (2009) identified 16704 landslides, and estimated that the number of earthquake landslides in Wenchuan earthquake reached 3.5 $\times$

10ˆ4. Daiet et al.( 2011) identified about 5.6 × 10ˆ4 landslides, Gorum et al. ( 2011) identified about 6 × 10ˆ4 landslides. Xu et al. (2014) interpreted the number of land-slides as 197 481. For Lushan earthquake, Cui Peng et al. (2013) identified about 1,460 landslides. Chang Ming et al (2013) identified 703 landslides. Pei (2013) esti-mated 1,800 geological hazard sites. Xu et al. (2015a, 2015b) identified the number of landslides in Lushan earthquake is 15645 and 22528 respectively. In addition, this paper studies coseismic landslides. We believe that the rainy season will cause new landslides, so the collected remote sensing data are mainly before the rainy season in China (June to September). The Wenchuan earthquake data were taken on May 30 and June 4, 2008, and the Wenchuan earthquake data were taken on April 20, 21 and May 13, 2013, which may result in less data than the existing data. As a representative of our group, I express my sincere appreciation here to editors and specialist reviewer for your instructions and helps to the article.

(References: Huang R Q, Li W L 2009.Analysis on the number and density of land-slides triggered by the 2008 Wenchuan earthquake, China. Journal of Geological Haz-ards and Environment Preservation,20( 3) : 1- 7. Dai F C, Xu C, Yao X, et al. 2011. Spatial distribution of landslides triggered by the 2008 MS 8. 0 Wenchuan earthquake, China .Journal of Asian Earth Sciences,40( 4) : 883- 895. Gorum T, Fan X M, Westen C J V, et al. 2011. Distribution pattern of earthquake induced landslides triggered by the 12 May 2008 Wenchuan earthquake. Geomorphology,133(3-4) : 152- 167. Xu C, Xu X W, Yao X, et al. 2014. Three( nearly) complete inventories of landslides trig-gered by the May 12,2008 Wenchuan MW7. 9 earthquake of China and their spatial distribution statistical analysis. Landslides,11( 3) : 441- 461. Cui P, Chen X Q, Zhang J Q, et al. 2013. Activities and tendency of mountain hazards induced by the MS7. 0 Lushan earthquake, April 20,2013. Journal of Mountain Science,31( 3) : 257- 265. Chang M, Tang C, Li W L, et al. 2013. Image interpretation and spatial analysis of geo-hazards induced by" 4.20" Lushan earthquake in epicenter area. Journa1 of Chengdu University of Technology ( Science & Technology Edition) ,40( 3) : 275- 281. Pei X J, Huang R Q. 2013. Analysis of characteristics of geological hazards by " 4.20" Lushan

earthquake in Sichuan,China.Journa1 of Chengdu University of Techno1ogy( Science ïijĘ Technology Edition) ,40( 3) : 257- 263. Xu C, Xu X W, Shyu J B H, et al.2015a. Landslides triggered by the 20 April 2013 Lushan, China, MW 6. 6 earthquake from field investigations and preliminary analyses. Landslides,12 ( 2 ) :365- 385. Xu C, Xu X W, Shyu J B H.2015b. Database and spatial distribution of landslides triggered by the Lushan, China MW 6. 6 earthquake of 20 April 2013. Geomorphology,248: 77- 92.)

**Fig. 1.** Yellow star means 2008 Wenchuan earthquake and red star means 013 Lushan earthquake

---

## Author Response (AR1)

We very much appreciate the careful reading of our manuscript and valuable suggestions of the reviewer. All comments are very important for my thesis writing and scientific research work. Below is our response to the issues raised in the review reports. Please find the changes in track-change version.

**Responds to the referee's comments:**

**Referee A**

**Comment 1:** The author considered that the scale frequency relationship of landslides developed in intensity VII-IX conforms to one law, that of landslides in intensity x conforms to another law, and that in intensity XI onforms to another law. First of all, there are very few landslides in VII and VIII, and there will be a great deal of uncertainty in the law obtained from such a small sample.

Response: In this paper, landslide data were accessed through by field investigation and interpretation of high-resolution remote sensing images. We got 108 landslides by field investigation and 706 by remote sensing interpretation in VII zone, 108 landslides by field investigation and 477 by remote sensing interpretation in VIII zone. The field survey data is small because we can only measure the landslides visible along the highway. In addition, the landslide data are divided into different seismic intensity zones, so the sample size in each zone seems to be small.

In the study of frequency-magnitude distribution of landslides, the sample number of different landslides databases are quite different. For example, Brunetti (2009) examined 19 landslide datasets. Individual datasets include from 17 to 1019 landslides of different types. Each landslide dataset exhibits heavy tailed (self-similar) behaviour for their frequency-size distributions.

 (**References:**Brunetti M., Guzzetti F., and Rossi M.: Probability distributions of landslide volumes. Nonlinear Processes in Geophysics, 16(2):179-188. https://doi.org/10.5194/npg-16-179-2009,2009.)

**Comment 2:** The ground motion accelerations used in the sand pile experiments carried out by the authors are 0.075-0.125g and 0.15-0.25g, although the phenomena claimed by the author are consistent with those obtained from the spatial distribution of landslides. However, 0.075-0.125g does not correspond to VII-IX intensity area. In fact, in general, 0.2g corresponds to the VIII degree region. Obviously, the boundary conditions of spatial analysis are completely different from those obtained by physical experiments, and it seems that there is no comparability.

Response: In the shaking table experiments, the one-side slope sand pile was built by dried sand gravel reaching its natural angle. The strength of the loose slope is less than the natural slope. In other words, a small peak acceleration could cause the sand pile to collapse in the experiment. Therefore, the seismic intensity in the physical experiment is less than in the real earthquake events. The mainly purpose of the experiment is to study the evolution trend of the distribution law with increasing seismic intensity, which is a comparison of the evolution law without a definite numerical correspondence.

Changes in text: We added an explanation of this issue, see lines 336-340.

**Comment 3:** The landslide data of different intensities used in this paper come from two earthquake events. Wenchuan earthquake is mainly high intensity area, while Lushan earthquake is mainly medium intensity area. Undoubtedly, the difference of different seismic characteristics and the nature of earthquake affected area, such as geological and topographical conditions, may greatly affect the results, so that they are not comparable.

Response: In general, landslides triggered by earthquakes are rare events, and the probability of two earthquakes occurring in the same area is even lower, providing valuable information for scientific research. Both the Wenchuan and Lushan earthquakes occurred within the Longmen Mountain fold-and-thrust belt. There is a gap extending about 75km between the main-shock of 2008 Wenchuan and Lushan events(Fig.1). Most scholars believe that the terrain and geological conditions are similar, and the thrust mechanism is similar. Some even suggest that Lushan earthquake was a strong aftershock after Wenchuan earthquake.

[Figure]

(*Yellow star: 2008 Wenchuan earthquake     Red star:2013 Lushan earthquake*)

In the study of landslide distribution, the distribution of landslides triggered by different evens (e.g., earthquake, rainfall) is often compared (Dussauge et al., 2003; Brunetti et al., 2009). In this paper, the landslide distribution is analyzed according to different seismic intensity, which is different from other similar studies. We believe that geological and topographical conditions may influence the parameters of the landslide distribution function, but the distribution form of the same seismic intensity area is the same.

Changes in text: We added a discussion of this issue, see lines 419-420.

 (**References:** *Dussauge, C., Grasso, J. R., and Helmstetter, A. Statistical analysis of rockfall volume distributions: implications for rockfall dynamics, J. Geophys. Res., 108(B6), 2286, doi:10.1029/2001JB000650, 2003.*

*Brunetti M., Guzzetti F., and Rossi M.: Probability distributions of landslide volumes. Nonlinear Processes in Geophysics, 16(2):179-188. https://doi.org/10.5194/npg-16-179-2009,2009.*)

**Comment 4:** The data quality of landslides triggered by the two earthquakes does not seem to be perfect. Compared with the existing work, either the whole earthquake area is not covered, or many landslides seem to be missed. There is no doubt that the law revealed by such data may deviate from the actual situation.

Response: Because of the difference of image precision, interpretation method and evaluation standard, the number of landslides triggered by Lushan earthquake and Wenchuan earthquake obtained by different experts is quite different.

Typical examples are:

For Wenchuan earthquake, Huang Runqiu et al. (2009) identified 16704 landslides, and estimated that the number of earthquake landslides in Wenchuan earthquake reached $3.5 \times 10^4$. Daiet et al.( 2011) identified about $5.6 \times 10^4$ landslides, Gorum et al. ( 2011) identified about $6 \times 10^4$ landslides. Xu et al. (2014) interpreted the number of landslides as 197 481.

For Lushan earthquake, Cui Peng et al. (2013) identified about 1,460 landslides. Chang Ming et al (2013) identified 703 landslides. Pei (2013) estimated 1,800 geological hazard sites. Xu et al. (2015a, 2015b) identified the number of landslides in Lushan earthquake is 15645 and 22528 respectively.

In addition, this paper studies coseismic landslides. We believe that the rainy season will cause new landslides, so the collected remote sensing data are mainly before the rainy season in China (June to September). The Wenchuan earthquake data were taken on May 30 and June 4, 2008, and the Wenchuan earthquake data were taken on April 20, 21 and May 13, 2013, which may result in less data than the existing data.

Changes in text: We added a discussion of this study, see lines 392-395.

(**References:** Huang R Q, Li W L 2009.Analysis on the number and density of landslides triggered by the 2008 Wenchuan earthquake, China. Journal of Geological Hazards and Environment Preservation,20( 3) : 1- 7.

Dai F C, Xu C, Yao X, et al. 2011. Spatial distribution of landslides triggered by the 2008 MS 8. 0 Wenchuan earthquake, China .Journal of Asian Earth Sciences,40( 4) : 883- 895.

Gorum T, Fan X M, Westen C J V, et al. 2011. Distribution pattern of earthquake induced landslides triggered by the 12 May 2008 Wenchuan earthquake. Geomorphology,133(3-4) : 152- 167.

Xu C, Xu X W, Yao X, et al. 2014. Three( nearly) complete inventories of landslides triggered by the May 12,2008 Wenchuan MW7. 9 earthquake of China and their spatial distribution statistical analysis. Landslides,11( 3) : 441- 461.

Cui P, Chen X Q, Zhang J Q, et al. 2013. Activities and tendency of mountain hazards induced by the MS7. 0 Lushan earthquake, April 20,2013. Journal of Mountain Science,31( 3) : 257- 265.

Chang M, Tang C, Li W L, et al. 2013. Image interpretation and spatial analysis of geohazards induced by" 4.20″ Lushan earthquake in epicenter area. Journa1 of Chengdu University of Technology ( Science & Technology Edition) ,40( 3) : 275- 281.

Pei X J, Huang R Q. 2013. Analysis of characteristics of geological hazards by " 4.20″ Lushan earthquake in Sichuan,China.Journa1 of Chengdu University of Techno1ogy( Science & Technology Edition) ,40( 3) : 257- 263.

Xu C, Xu X W, Shyu J B H, et al.2015a. Landslides triggered by the 20 April 2013 Lushan, China, MW 6. 6 earthquake from field investigations and preliminary analyses. Landslides,12 ( 2 ) :365-

385.

*Xu C, Xu X W, Shyu J B H.2015b. Database and spatial distribution of landslides triggered by the Lushan, China MW 6. 6 earthquake of 20 April 2013. Geomorphology,248: 77- 92.*)

**Referee B**

**Comment 1:** From the title and the abstract, I thought the objective of the paper was to find the distribution law between size and frequency of landslides triggered by earthquake with different intensities. However, the paper structure should be well managed considering three ways (inventory data, computer simulation and physical experiment) in the paper.

**Response:** We thank the reviewer for the suggestion.

Changes in text: See lines 86,236 and 327.

**Comment 2:** Landslide inventory data in the paper is not clearly clarified. The process and existing uncertainties in image interpretation for landslides should be explained. How did the authors deal with the connected landslides which are common in Wenchuan earthquake events and difficult to be separated? The author states the phenomena in lines 160-161, but without any other words later.

**Response:** Several individual landslides in low-resolution images may be misjudged into a single larger landslide because the distances of them are less than the resolution of the image. Therefore, the higher resolution of the images used, the easier individual landslides are separated and the more objective landslide inventory maps prepared. The remote sensing data collected in this paper have high resolution, and the single landslide can be identified through manual visual inspection combined with experience.
Changes in text: We added an explanation of this issue, see lines 130-136.

**Comment 3:** The distribution law of volume (depth)-frequency and area-frequency is obtained and shown in Table 1 and Table 2. The results are from the same triggering events but the number of the samples shown in the tables is not the same. Why?

**Response:** The samples in Table 1 are obtained from field surveys, and the samples in Table 2 are from remote sensing interpretation. The number of samples obtained from field surveys is small, while that from remote sensing interpretation is large.
Changes in text: We emphasized the data sources in Table 1 and Table 2,see lines 159 and 195.

**Comment 4:** Equation 1 is wrong.

**Response:** We are very sorry for our incorrect writing, I have revised it.

The correct equation is:

$$N(A) = a \times A^{-b}$$

Changes in text: See Line 150.

**Comment 5:** Please explain the matching ability of the physical experiment with the real earthquake events, such as in terms of the peak acceleration in experiment and seismic intensity in Table 2-3.

**Response:** In general, 0.1g corresponds to the VII degree region, 0.2g corresponds to the VIII degree region, 0.4g corresponds to the IX degree region. In the physical experiment, 0.075-0.125g corresponds to the region blow IX degree, and 0.15-0.25g corresponds to the region above IX degree. In the shaking table experiments, the one-side slope sand pile was built by dried sand gravel reaching its natural angle. The strength of the loose slope is less than the natural slope. In other words, a small peak acceleration could cause the sand pile to collapse in the experiment. Therefore, the seismic intensity in the physical experiment is less than in the real earthquake events. The mainly purpose of the experiment is to study the evolution trend of the distribution law with increasing seismic intensity, which is a comparison of the evolution trend without a definite numerical correspondence.
Changes in text: We added an explanation of this issue, see lines 336-340.

**Technical corrections** 1. Grammar mistake exists in the paper, such as Lines 30-34, Line 211. 2. The quality of the figures need to be improved. 3. The unit of the parameters in Line 234 and Figure 4-5 is not clear. 4. The language needs to be improved.

**Response:** Thank you for pointing out the mistakes in our writing, we have corrected.
The revised details can be found in Line 33, Line 285. The unclear parameters have been revised in line 286.
The parameters in line 234 and Figure 4-5 are dimensionless.
Changes in text: We added a description of the parameter unit and revised the title of Figure 4, 5, see lines 309,234 and 235.

**In the end, as a representative of our group, I express my sincere appreciation here to editors and specialist reviewer for your instructions and helps to the article.**